# Defining Coral Bleaching as a Microbial Dysbiosis within the Coral Holobiont

**DOI:** 10.3390/microorganisms8111682

**Published:** 2020-10-29

**Authors:** Aurélie Boilard, Caroline E. Dubé, Cécile Gruet, Alexandre Mercière, Alejandra Hernandez-Agreda, Nicolas Derome

**Affiliations:** 1Institut de Biologie Intégrative et des Systèmes (IBIS), Université Laval, Québec City, QC G1V 0A6, Canada; aurelie.boilard.1@ulaval.ca (A.B.); cecile.gruet.1@ulaval.ca (C.G.); 2California Academy of Sciences, 55 Music Concourse Drive, San Francisco, CA 94118, USA; alejhernandez@calacademy.org; 3PSL Research University: EPHE-UPVD-CNRS, USR 3278 CRIOBE, Université de Perpignan, 66860 Perpignan CEDEX, France; alexandre.merciere@gmail.com; 4Laboratoire d’Excellence “CORAIL”, 98729 Papetoai, Moorea, French Polynesia; 5Département de Biologie, Faculté des Sciences et de Génie, Université Laval, Québec City, QC G1V 0A6, Canada

**Keywords:** adaptive dysbiosis hypothesis, coral bleaching, holobiont, microbiota, Symbiodiniaceae

## Abstract

Coral microbiomes are critical to holobiont health and functioning, but the stability of host–microbial interactions is fragile, easily shifting from eubiosis to dysbiosis. The heat-induced breakdown of the symbiosis between the host and its dinoflagellate algae (that is, “bleaching”), is one of the most devastating outcomes for reef ecosystems. Yet, bleaching tolerance has been observed in some coral species. This review provides an overview of the holobiont’s diversity, explores coral thermal tolerance in relation to their associated microorganisms, discusses the hypothesis of adaptive dysbiosis as a mechanism of environmental adaptation, mentions potential solutions to mitigate bleaching, and suggests new research avenues. More specifically, we define coral bleaching as the succession of three holobiont stages, where the microbiota can (i) maintain essential functions for holobiont homeostasis during stress and/or (ii) act as a buffer to mitigate bleaching by favoring the recruitment of thermally tolerant Symbiodiniaceae species (adaptive dysbiosis), and where (iii) environmental stressors exceed the buffering capacity of both microbial and dinoflagellate partners leading to coral death.

## 1. Introduction

Coral reefs are one of the most biologically diverse and economically important marine ecosystems on the planet. Often called the rainforest of the sea due to their outstanding biodiversity, coral reefs only cover less than 0.1% of the ocean seafloor [1,2,3]. Coral reefs thrive in oligotrophic waters [4,5,6], yet they harbor more than 25% of all marine species [7,8], including about 30% of all marine fish species [1]. This ecosystem also delivers key services such as fisheries [9,10], tourism-based industries [11], coastal protection [12], and medicines [13,14,15], sustaining the welfare and livelihoods of millions of people. Coral reefs constitute complex ecosystems. Like trees in a forest, corals are foundational species responsible for creating structural complexity [16] and they are critical players in the nutrient recycling on reefs [17]. Corals are meta-organisms and are formed by a dynamic multipartite relationship between the cnidarian host, its endosymbiotic dinoflagellate algae (family Symbiodiniaceae [18]), and a suite of other non-Symbiodiniaceae microbes [19], hereafter termed the microbiota or microbes. The microbiota includes prokaryotes (archaea and bacteria) [20], eukaryotes (fungi and non-Symbiodiniaceae protists), and viruses [21,22]. In the present review, the dinoflagellate algae and microbiota are collectively termed the coral microbiome. Each partner plays a fundamental role within the holobiont. The coral host provides the living space and well-protected habitat for all partners involved, as well as a supply of nutrients and other metabolic byproducts [23]. The dinoflagellate endosymbionts deliver oxygen and organic compounds to their hosts, mostly via the transfer of photosynthetically fixed carbon [24,25], while the microbiota provides multiple beneficial functions, such as protection against pathogens, nitrogen fixation [26,27], sulfur cycling [28,29,30], and supply of other micronutrients [31,32]. The hologenome and the host–Symbiodiniaceae–microbiota interaction drive the biology of the holobiont [22,33] and ultimately define its phenotype [31].

The stability of the holobiont is fragile. Holobionts can transition from eubiosis (healthy state of the holobiont) to dysbiosis (unhealthy, disrupted state of the holobiont) as, for instance, environmental conditions deteriorate. The health of the holobiont depends on many biotic (pathogens, prey availability, cnidarian host physiology and genetic background, assemblage of photosynthetic algae and microbes, among others) and abiotic (temperature, irradiance, pH, water movement, nutrients, among others [34]) factors, some of which are directly or indirectly impacting the holobiont homeostasis. The rapid pace of environmental change caused by climate warming and other anthropogenic stressors is now posing a serious threat to coral reef ecosystems and has been linked to numerous occurrences of coral holobiont dysbiosis [31,35], causing higher susceptibility to opportunistic pathogens and, ultimately, coral mortality. Of all the natural and anthropogenic stressors imposed on coral reefs, increasing temperature is one of the most imminent threats as it can act from a few hours (i.e., pulse warming events) to a few years [36,37]. The main consequence of these climate-driven marine heat waves is coral bleaching. Coral bleaching is the result of the breakdown of the obligate symbiosis between the coral animal and its photosynthetic dinoflagellate endosymbionts [38], leading to the white calcium carbonate skeleton being visible through the transparent host tissue (Figure 1). This dysbiosic state can occur either when the host immune system activates cell apoptosis, autophagy, exocytosis, detachment or necrosis pathways [39], and rejects the dinoflagellate partner, or when the pigments are directly expulsed from the algae when their thylakoids are exposed to free oxygen radicals [38,40]. Although evidence suggests that the production of reactive oxygen species (ROS) in the Symbiodiniaceae cells is the most likely cause of their expulsion [41], the bleaching mechanism is still far from being fully understood [42].

Since the first description in the 1980s [43], increases in sea surface temperature have triggered several unprecedented mass coral bleaching events, including three global-scale bleaching events in 1998, 2010, and 2016 [44]. The frequency and severity with which coral bleaching occurs has increased in recent years [45,46]. Hence, the survival and persistence of corals will depend on their ability to increase their thermal tolerance at a rate that keeps pace with global warming [47]. If not, over 70% of all coral reefs are expected to disappear by the end of 2050 [48,49]. The severity of coral bleaching depends on several factors, including the specific coral species impacted, the dinoflagellate and bacterial community composition, and the thermal history of the holobiont [50,51]. Nevertheless, key microbial associates have been identified as a potential source of adaptive variation in response to changing environmental conditions [31,52], in addition to their role in reducing the presence of pathogens [53]. Given the key roles exerted by the microbiota and the Symbiodiniaceae in maintaining the host’s health and homeostasis [54,55,56], numerous studies have investigated the mechanisms by which these symbiotic partners can mitigate coral bleaching and increase resilience of the coral holobiont to other climate related stressors [36,57,58]. This review aims to explore the resilience to dysbiosis and bleaching in hermatypic reef-building corals (Scleractinia, Hexacorallia). This will be achieved through a three-step approach: we will (i) briefly portray the great taxonomic diversity of the coral holobiont, (ii) explore coral resilience to bleaching events and propose the hypothesis of adaptive dysbiosis as a mechanism of environmental adaptation, and (iii) mention potential solutions to mitigate coral bleaching and suggest new research avenues.

## 2. The Coral Holobiont

The coral holobiont comprises the cnidarian host and all its associated microscopic organisms that functionally interact with one another via several metabolic pathways, further contributing to the physiology and health of the holobiont. The coral holobiont can be separated into three main components: (i) the cnidarian host, (ii) the Symbiodiniaceae algae, and (iii) the microbiota [59].

### 2.1. Corals

Since their first appearance around 425 million years ago, scleractinian corals have radiated into more than 1500 species [60], about 900 of which are hermatypic corals normally containing millions of endosymbiotic dinoflagellate algae in their tissues [61]. Reef-building corals (Cnidaria, Anthozoa, Scleractinia) are invertebrate, modular organisms composed of many identical units, called polyps, connected by an interlaying tissue, the coenosarc [61,62,63]. The polyp is composed of two cell layers, the ectodermis and gastrodermis separated by the mesoglea, and each polyp has an oral disk and a mouth surrounded by small tentacles, corresponding to the opening of the gastrovascular cavity (Figure 2). The coral skeleton is secreted by specialized calcifying cells in the calicodermis [64,65,66], while the calcification process occurs in the calcifying fluid that is formed between the calicodermis and the skeleton [64] (Figure 2). The cellular glands, on the other hand, secrete a mucus layer to protect the ectodermis [67]. The ectodermis, gastrodermis, gastric cavity, mucus layer, and skeleton correspond to distinct microhabitats within the coral host that can encompass functionally distinct microbial communities [68,69]. These microhabitats change dramatically (in structure and composition) during bleaching periods. For example, depending on the bleaching stage and intensity, the loss of symbiotic algae leads to distinct degrees of tissue degradation [39,70,71]. Compared to healthy corals, the mucus surface layer in bleached corals is thinner [72] with a distinctive sugar composition [73]. Bleaching-associated changes also represent a modification of the microorganisms’ habitats.

As in other invertebrates, an innate immune system allows corals to maintain homeostasis. Invertebrates lack an adaptive immune system [74], but their genomic repertoire is elaborated [75,76]. The immediate reaction via recognition, signaling pathways towards activation of defense mechanism, and effector responses (reviewed in [77]) allows the detection and management of potential threats to host integrity. Specific mechanisms of the coral immune response are still poorly understood. However, research in other cnidarians suggests that immunological processes occurring within the host’s microhabitats have a critical role in assembling the holobiont [76,78,79]. During bleaching, immune regulation activity is observed in apoptosis, apoptosis regulation, necrosis, and cell–cell adhesion proteins genes [80]. Heat-tolerant corals have a higher expression of genes related to cell death signaling and immunity (Hsp gene family), evidencing the frontload expression of the host immune system to maintain physiological resilience during environmental stress [80]. However, the coral immune response to high temperature and eventual susceptibility to diseases after heat stress seems to be species-specific [81]. Importantly, gene expression has been found uncorrelated or incongruent with the concentration of the encoded proteins [82,83,84], pointing out that conclusions drawn from the exclusive use of mRNA concentration data should be cautiously interpreted.

Corals can reproduce through both sexually and asexually (including budding, fragmentation, polyp bailout, asexual planula larvae, and embryo breakage) [61,85]. Corals can be hermaphroditic or gonochoric and they exhibit two modes of sexual reproduction, namely broadcast spawning and brooding. During broadcast spawning, the coral releases eggs and sperm into the water column, where fertilization, embryo, and larval development occurs. During brooding, a coral colony releases sperm that fertilizes the eggs that are inside another colony, and the larva is internally brooded and released into the water column once maturity is reached. Host genetic information inherited from parents may provide a source of adaptive variation in corals, such as epigenetic marks and somatic mutations. Coral adaptation based on standing genetic variation has also been shown, where natural variation in heat tolerance (among other forms of stress) is heritable and evolvable [86,87]. Nevertheless, the capacity for host genome adaptation to warming might be limited because of the long generation times associated with most coral species (~4–8 years) [88,89].

The reproduction mode influences the early establishment of the microbial consortium in corals. The Symbiodiniaceae endosymbiont transmission is generally horizontal in broadcast spawners, and vertical and mixed in brooders [93,94,95]. However, the establishment is not correlated to the reproduction mode or not well understood for other microbial community members (bacteria, viruses, fungi). Particularly for bacterial communities, the role of maternally inherited and environmentally acquired microbes on the community assembly remains unclear, where both broadcast spawners and brooders can inherit bacteria via vertical transmission [95,96,97]. However, regardless of the reproduction mode, the acquisition of bacteria via horizontal transmission seems to have a critical role in winnowing the bacterial community as the coral matures [95,98]. Transgenerational plasticity by conditioning maternal colonies before transmission and manipulating the horizontally acquired microorganisms has been proposed as a potential pathway to facilitate coral acclimatization and adaptation [99].

### 2.2. Symbiodiniaceae

Zooxanthellae are crucial components of the coral holobiont. The endosymbiotic dinoflagellates, from the family Symbiodiniaceae (order Suessiales, class Dinophyceae [18]) are photosynthetic partners living in vacuoles (the symbiosomes [100]) in the gastrodermis of the coral host (Figure 2). Corals mostly depend on the dinoflagellate symbionts for energy requirements through the translocation of photosynthates produced via photosynthesis, including glucose, glycerol, and amino acids [63,101,102]. Through photosynthesis, the Symbiodiniaceae algae also produce the oxygen required for coral host cellular respiration and free oxygen radicals that can act as a protection against pathogenic infections [103]. In exchange, the dinoflagellates benefit from this partnership by being protected against predators, living in a stable and light-rich environment with direct access to inorganic waste produced by the host [19,24]. Corals provide nitrogen under the form of nitrates (NO_3_^−^) to the algal symbionts [27], while the Symbiodiniaceae use carbonic gas (CO_2_^−^) produced by the polyps, which also promotes the dissociation of carbonic acid (H_2_CO_3_) from the water column and allows calcium carbonate (CaCO_3_) to precipitate. Thus, this photosymbiosis also powers light-enhanced calcification and the rapid growth of corals [104,105,106], resulting in coral reef development [107].

Based on the phylogenetic disparity in the Internal Transcribed Spacer 2 (ITS2) sequence, these dinoflagellate algae were initially distinguished into nine phylogenetic clades (nominated A to I) within the genus *Symbiodinium* [19,108,109]. The family Symbiodiniaceae was recently revised and reorganized in seven genera: *Symbiodinium*, *Breviolum*, *Cladocopium*, *Durusdinium*, *Effrenium*, *Fugacium*, and *Gerakladium* (formerly clade A to G, respectively) [18]. Among them, four genera are known to commonly associate with scleractinian corals, *Symbiodinium*, *Breviolum*, *Cladocopium*, and *Durusdinium*, while the genera *Fugacium* and *Gerakladium* are only occasional scleractinian associates [110,111]. Coral-associated Symbiodiniaceae communities can involve one or more species of algae, where the species makeup can be attributed to specific physiological and biochemical properties, often associated to their adaptation to distinct environmental conditions (e.g., carbon availability, temperature, and pH) [101,112,113]. The host–symbiont specificity can also be attributed to the coral host phylogeny and habitats. Host-specific Symbiodiniaceae associations have been described over a large depth gradient for some coral species in the Red Sea [110] and Caribbean [114], while variations in Symbiodiniaceae associations between and within coral species are commonly attributed to depth-mediated gradients of light and temperature [115,116]. For instance, the photosynthetic properties of *Symbiodinium*, *Breviolum*, and *Durusdinium* species are more suited for light regimes associated with shallow habitats [117,118], with some *Durusdinium* species being the most thermally tolerant [110,119]. On the other hand, *Cladocopium* species are considered depth-generalists, although populations of shallow habitats can exhibit greater assemblage diversity compared to deeper habitats at some reef locations [120].

A possible determinant of host-specificity and environmentally induced variations in Symbiodiniaceae composition is the mode of symbiont acquisition. The Symbiodiniaceae algae can be passed on to the offspring vertically (vertical transmission) [94,121,122]. Vertical symbiont transmission creates heritability in Symbiodiniaceae community composition, often resulting in tight co-evolution between the host and its microalgae, which may limit the host flexibility to associate with taxonomically and physiologically diverse dinoflagellate partners [121]. Such a mode of transmission promotes the evolution of specialist symbionts that are more locally adapted to their native environment [121,123]. On the other hand, symbiotic-free gametes, or larvae, acquire passively the Symbiodiniaceae cells from the surrounding environment (horizontal transmission) [63]. Horizontal symbiont transmission can potentially lead to new host–Symbiodiniaceae associations relative to the maternal colony, conferring a higher flexibility on the host to associate with a wide range of symbionts. By acquiring new symbiont partners during their early life history, horizontally transmitting corals are considered as generalists, and this flexibility may be advantageous in terms of range expansions [124], adaptation to changing environmental conditions, and stress tolerance [125]. Consistently, the dispersal potential of offspring is higher when horizontal symbiont transmission results in a diverse Symbiodiniaceae assemblage, providing them with better abilities to adapt and survive to a wider range of new environments [126,127]. Regardless of the symbiont transmission mode, the coral host has the ability to regulate the density and diversity of their associated Symbiodiniaceae during its lifetime [27,128]. This flexibility may occur via three different cellular mechanisms: (i) variation of coral tissue compounds, (ii) deterioration or digestion of Symbiodiniaceae, and (iii) limiting access to nutrients (e.g., nitrogen and phosphorus) [27].

### 2.3. The Microbiota

In scleractinian corals, the microbiota encompasses a wide range of microbial organisms, including archaea, bacteria, eukaryotes (endolithic algae, fungi, protists), and viruses [19,22,129], all of which play key roles in regulating the physiology of the holobiont. The microbiota serves diverse functions in the coral holobiont, including nutrient acquisition, protection against pathogens, immune system conditioning, and stress tolerance [32,130]. The microbiota also ensures nitrogen fixation and sulfur cycling [20], as well as a continuous supply of other critical nutrients, including phosphorus, metals, and vitamins [31,32]. The microbiota activity can also help in adjusting the host physiological responses to stress and contribute indirectly to the photosynthetic activity via nitrogen- and sulfur-cycling microbes [27,131]. The coral host in turn provides several niches (tissue, skeleton, and mucus, Figure 2) for the microbes, as well as nutrients and carbon [31]. Microbiota members exhibit a wide spectrum of symbiotic interactions with their host, ranging from mutualism to commensalism and parasitism [31]. The number of genes with unique functions can also surpass that of the host [33,132]. Consequently, changes in terms of the microbial taxonomic composition and functional repertoires (i.e., the metagenome) represent an evolutionary process in action that could potentially produce heritable adaptive phenotypic traits [33,132]. Although the cnidarian genome remains unchanged, the coral holobiont is continuously evolving.

The heritability of such phenotypic traits can result partially from vertical transmission of the microbiota, i.e., when the host passes on a sample of its microbes to the offspring [133]. This vertical transmission creates heritability in microbial compositions, allowing hosts to transfer specific and successful microbial species to their progeny [134]. Although microbes can be transmitted both vertically and horizontally in corals [95,96,97,135,136,137,138], mixed strategies also exist [31,98,134]. In broadcast spawning corals, one example is the release of specific bacteria from the coral host into the immediate environment, favoring the recruitment of those species by the larvae [139]. Alternatively, microbial transmission can also occur through their incorporation into the mucus that surrounds egg–sperm bundles [96,140]. However, the heritable association between a host and its microbes is weakened over time because most coral species continuously acquire microbial symbionts from their surrounding environment [98,141]. The microbiota assembly could have an initial stage with a maternal-inherited community that reaches maturity through successional stages ruled by horizontal acquisition, with the potentiality to generate functional redundancy within the microbiome [79] and to grant thermal tolerance to the coral host [130].

Overall, the coral host represents a great diversity of microhabitats (gastrovascular cavity, tissue, skeleton, and mucus layer), each sheltering a specific set of microbes [22,32,142]. Each of these coral compartments is characterized by a specific set of microbes and associated functional repertoires, emphasizing the important effects of microhabitat partitioning on coral microbiome structure and function [20]. In both tissue layers (ectodermis and gastrodermis) of the polyp, cell-associated bacteria are often found in aggregates, called cell-associated microbial aggregates (CAMAs), that most likely consist of a single bacterial species [92,143,144,145] (Figure 2). The endolithic community refers to all microorganisms living in the coral skeleton. The skeleton houses the highest diversity of bacteria relative to other microhabitats, in addition to fungi and green algae [146,147,148]. Most coral-associated bacteria reside in the surface mucus layer covering the ectodermis of the polyp [149], which is a nutrient-rich habitat with a high amount of dissolved organic and inorganic compounds (e.g., carbon and phosphorus) [150,151]. Mucus-associated bacterial community compositions are mostly determined by two main mechanisms: (i) the composition (e.g., nutrients, biocides, signals), viscosity, and thickness of the mucus [67,152], and (ii) the specific receptors associated with the coral surface and mucus layer [153]. The mucus could serve as a means to discriminate beneficial microorganisms from pathogens [154] by preventing pathogens’ antibiotic production, necessary for bacterial colonization [53], or by modifying the nutritive and physical characteristics of this microhabitat. The highly diversified mucus-associated bacterial communities observed in hosts harboring diverse Symbiodiniaceae genera might result from distinct Symbiodiniaceae-derived nutrient compositions of the mucus [155].

#### 2.3.1. Bacteria

Bacterial symbionts are key players supporting the functioning and health of corals and reef ecosystems [31]. Corals can develop and maintain some level of specificity with their associated bacterial symbionts [63,156,157], suggesting an adaptive co-evolution between the coral host and its microbiota [158]. Species-specific microbiota could correlate with the phylogeny of their coral host (i.e., phylosymbiosis) [158,159,160]. The environment also plays a significant role in structuring coral-associated bacterial communities, where a coral species can have very different bacterial compositions depending on its habitat, pointing to an adaptation to local environmental conditions [21,161,162,163]. The microbiota associated with reef corals has been reported as one of the most diverse [63], spanning 37 described bacterial phyla [164]. Bacterial communities of scleractinian corals are dominated by Proteobacteria [21] (mostly of the orders Alpha- and Gammaproteobacteria), Bacteroides [164], Cyanobacteria [21], and Firmicutes [164]. The *Endozoicomonas, Vibrio*, and *Serratia* genera are the most represented [164], with bacterial species showing a high level of specificity with their coral host [63]. These bacterial symbionts play a myriad of key functions in the holobiont, including nutrient acquisition and cycling (carbon, nitrogen, sulfur, and other metabolites) [20], oxygen radical [35,165] and metal degradation [20], control of Symbiodiniaceae growth, density, and nutrition [166,167,168], protection against pathogens [169,170], and stress tolerance [27,130]. For example, functional genes associated to the bacterial community of *Orbicella faveolata* and *Porites lutea* are involved in nutrient cycling (carbon, nitrogen, sulfur), featuring genes for carbon fixation (through the Calvin cycle, reductive acetyl-CoA pathway, and reverse Krebs cycle) and degradation [20,132]. Interestingly, genes encoding mechanisms to establish stable symbiosis with the host (ankyrin repeats, ARPs; and WD40-containing proteins) were identified across bacterial phyla associated with *P. lutea* [132].

When the availability of nitrogen decreases in the environment, bacteria can supply the host with organic nitrogen. The microbiota harbors nitrifying (transforms ammonium into nitrates and eventually nitrites) and denitrifying (reduces nitrites and nitrates into dinitrogen) bacterial species that transform nitrogen-related waste produced by the host into non-toxic compounds [20,27]. By regulating the nitrogen cycle, these bacteria can prevent accumulation of nitrogenous waste in some compartments that could alter coral homeostasis. These bacteria can also regulate dinoflagellate densities by maintaining a balanced ratio of nitrogen and phosphate via the utilization of ammonium (NH_4_^+^) [128]. The ammonium produced by the coral host [27] is the preferred inorganic nitrogen source of the Symbiodiniaceae [171], while the utilization of nitrate (NO_3_^−^) reduces photosynthetic health [172]. The occurrence of anaerobic ammonium oxidation (anammox) from the coexistence of nitrifying and denitrifying bacteria (conversion of ammonium and nitrates into dinitrogen) thus constrains the dinoflagellate algae to use dinitrogen (N_2_) for photosynthesis, which can ultimately increase Symbiodiniaceae abundance, promote photosynthetic health, and enhance coral health [128,172]. Because the dinoflagellates produce chemical compounds that favor bacterial development and growth, such an increased density of Symbiodiniaceae cells may have an impact on the bacterial composition through syntrophy (i.e., cross-feeding) [173,174]. Diazotrophic communities (cyanobacteria and bacteria) also play a key role in nitrogen cycling by providing diazotrophically derived nitrogen (DDN) to the host and photosynthetic symbionts [175]. *Endozoicomonas* symbionts of the bacterial family Hahellaceae also play a significant role in nutrient acquisition and cycling of organic compounds because of their ability to metabolize dimethylsulfoniopropionate (DMSP) [30], a molecule produced by the Symbiodiniaceae to mitigate osmotic stress. These bacteria transform the DMSP into dimethylsulfide (DMS) that reacts with toxic compounds and eliminates them [165]. Some coral-associated bacterial species express orthologous genes for antioxidant enzymes, bringing another protective mechanism against oxygen free radicals [35].

Another key role of the microbiota is to provide protection against pathogens via diverse mechanisms, including competitive exclusion, antibiotic production, inhibition of quorum sensing, and secondary metabolite production [176,177]. The coral mucus consists of polymeric glycoproteins and lipids [178] operating as a defense barrier to protect the coral against invasive pathogens [169]. Both commensals and pathogens living on the surface mucus layer employ glycosidases to utilize carbon and nitrogen sources within the coral mucus. However, commensal bacteria can disrupt the establishment of invading pathogens within the coral mucus by interfering with their metabolic activities and by swarming [179]. This exclusion of pathogenic bacteria by out-competing commensal bacteria (amensalism) can ultimately reduce the occurrence of coral diseases. Coral-associated bacteria can also produce a wide range of antimicrobials and biocides to protect the host against pathogens, notably pathogenic fungi [170]. The high concentration of antibiotics within the mucus layer prevents new pathogens from settling on the coral surface. Although considered as an efficient mechanism to keep invaders at bay, pathogens also produce antibiotics to favor their settlement on corals [53]. Quorum sensing is another molecular mechanism involved in the protection against pathogens, where the production of autoinducers used for chemical signaling [180] allows bacteria to modulate their gene expression depending on the cell population density [181]. Both gram-positive and gram-negative bacteria use these signals to regulate their physiological activities, such as symbiosis, virulence, motility, and antibiotic production [182]. Of note is the inhibition of bacterial quorum sensing for the prevention of infection in other hosts, such as marine algae [183].

Although heat-induced bleaching is triggered by the breakdown of the symbiosis between corals and their dinoflagellate symbionts, bleaching is sometimes accompanied by changes in the coral-associated bacterial community composition, ranging from mutualistic species to commensalistic, and parasitic [35,57,184]. Bleached corals are often associated with an increase in community diversity, while a reduction in mutualistic and key bacterial symbionts, such as *Endozoicomonas* species, has been observed [185]. On the other hand, shifts towards an increase in opportunistic bacteria and potential pathogens, such as *Vibrio* species [51,186,187], have been reported and associated with an increase in genes related to virulence factors [51]. Bourne et al. [186] also identified an increase in *Vibrio* spp. (about 0 to 30% of the relative abundance) in corals experiencing heat-induced bleaching, while their abundances return to normal in resilient corals (about 0% after the thermal stress). Although this study suggests that *Vibrio* spp. might be specifically linked to coral bleaching, current knowledge makes it impossible to disentangle whether their presence, increase in abundance, and virulence in thermally stressed corals is a consequence of bleaching [188] or plays an active role in the development of bleaching [189].

#### 2.3.2. Viruses

Viruses are found in both tissue and mucus layers, where they hold diverse essential functions in the coral holobiont, including bacterial population maintenance, protection against pathogens, nutrient cycling, and horizontal gene transfer [190,191]. As with bacteria, the surface mucus layer encompasses the highest abundance of viruses [192,193], suggesting that viruses may potentially exert a tight control on mucus-associated bacterial populations [194]. Coral-associated viruses are taxonomically diverse, spanning 60 different families [195], while being dominated by the Siphoviridae, Myoviridae, and Podoviridae families from the Caudovirales order [196]. Considering the large amount of available bacterial hosts within the coral holobiont, bacteriophages (viruses infecting bacteria) are a major microbial component [195], including nucleocytoplasmic large DNA viruses (NCDLVs), particularly those of the Phycodnaviridae and Mimiviridae families, as well as Poxviridae, Ascoviridae, and Iridioviridae [191]. While changes in the coral virome have been reported in different coral health states (healthy, diseased, bleached), nine to twelve families with various genome types (dsDNA, ssDNA, and retrotranscribing RNA) have been identified as core virome members [190].

Dominated by bacteriophages [195], the virome seems to mainly control and regulate bacterial populations through lysis [194] with a daily viral-mediated bacterial turnover ranging from 20 to 120% [190]. Mucus-associated phages can also act as a “lytic barrier” against bacterial pathogens and, as such, are considered an active component of the coral innate immune system [194]. Observations of binding between the phage capsid and the glycoprotein of the mucus layer have led to the bacteriophage-adhering to mucus model (BAM), which suggests a tight coral-phage co-evolution that limits the colonization by bacterial pathogens [197]. The model also includes the benefits of lysogeny, where phage-carrying (lysogenic) bacteria of the coral microbiome become resistant to lysis by other viruses [194,197]. Similar to their roles in microbial food-webs in oceanic ecosystems [198,199], viruses participate in nutrient acquisition and cycling within the holobiont by controlling the microbial biomass turnover [200]. They also contribute in the exchange of microbial genetic material via horizontal gene transfer, including transduction (among other mechanisms [190]), and may therefore act as “gene reservoirs”.

Rising temperature can also increase virulence by expressing genes causing dysbiosis (but see [82,83,84] for incongruence between gene expression and encoded proteins). Virulence factors can be acquired via prophages and/or horizontal transmission, which may further increase the virulence of some bacterial pathogens (e.g., *V. coralliilyticus*) [201]. Consequently, the distinction between viral causative agents and opportunistic strains, as for any other pathogens, remains a challenging task, mostly due to the presence of viruses in both healthy and diseased corals [51], and their complex interactions with other microbial members.

#### 2.3.3. Other Microbial Members

The microbiota also includes many other microbes such as archaea, protists, and fungi. However, much less attention has been paid to these microbial members and to their role in coral bleaching. Archaea are very common in marine environments. The archaeal groups Euryarchaeota and Thaumarcheota can associate with corals [132], with the latter being the most abundant and involved in nitrogen cycling [142,164]. Unicellular protists, the corallicolids (phylum Apicomplexa), have recently been identified as an important coral endosymbiotic partner and are highly abundant in healthy corals, along with ciliates and dinoflagellates [202]. The ARL-V is the most abundant group of corallicolids within the coral holobiont, and their localization in coral tissues differs from the endosymbiotic Symbiodiniaceae protist [202], while the photosynthetic activity of corals does not seem to be associated with their occurrence. Other protists have also been reported in corals, such as Licnophora and Coccidia that associate with *Pocillopora damicornis* [203].

Although there is growing interest in the endolithic communities, their functional role within the coral holobiont and bleaching responses remains poorly understood. The skeleton of many coral species is often dominated by the filamentous *Ostreobium* microalgae (Siphonales, Chlorophyta) [204] that seems to have co-evolved with both the coral host and Symbiodiniaceae symbiont [205]. Despite their significant role in reef erosion under elevated sea temperature and decreased pH [147,206,207,208,209], *Ostreobium* can potentially provide an alternative source of energy to the coral host during bleaching via the transfer of photosynthates [147,210]. Although coral-associated fungi were previously deemed as potential pathogens, little is known about their taxonomic and functional diversity within the endolithic community [211]. Yet, the fungal community associated with the skeleton of *Acropora hyacinthus* comprised a high diversity of basidiomycetes and ascomycetes species [129]. Fungi potentially contribute to nitrogen cycling [212], stress tolerance [213], and bioerosion reduction by parasitizing endolithic algae under environmental stress [213].

## 3. Coral Bleaching: From Adaptive to Traumatic Holobiont Dysbiosis

Since the partnership with microbial symbionts serves a wide range of beneficial functions within the coral holobiont, functional activity of microbes may mitigate or even prevent bleaching responses by supporting the internal equilibrium between the host and Symbiodiniaceae, and by maintaining stable rates of photosynthesis [27,128,134,166,172]. Conversely, a microbial dysbiosis can disrupt the stability of the photosynthetic activity and finely-tuned nutrient exchange between corals and their dinoflagellate partners [51,172], which can ultimately trigger the breakdown of this photosymbiosis, causing bleaching. Under temperature anomalies, coral health may therefore depend on the physiological changes within the holobiont that confer thermal tolerance, as well as its ability to return to a healthy equilibrium (i.e., resilience) after temperature returns to normal. At the ecosystem level, resilience is defined as the capacity of a system to absorb or withstand environmental stressors, and to build an alternative state, such that the system maintains its structure and functions [214,215]. At the coral holobiont level, resilience is defined as the capacity of the animal host to resist cellular stress under relatively extreme environmental conditions (averting coral death) or to revert back from cellular damages [216]. However, holobiont resilience also refers to the capacity to recruit new dinoflagellate symbionts and quickly reinstate the microbe’s functional activity to a healthy equilibrium after stress exposure.

As of today, the role of coral-associated microorganisms in bleaching remains poorly understood, although changes in bacterial community composition are frequently associated with coral bleaching [57,186,217]. These alternative microbial states (dysbiosis), and resulting shifts in the holobiont metabolic network, can occur very rapidly, and even before the first visual signs of bleaching [186]. As such, microbial profiling of corals, among other reef organisms, may confer significant advantages over traditional reef monitoring methods (usually based on visual signs of deterioration such as bleaching), and could therefore provide an early diagnostic tool to assess climate-related pressures on coral integrity [218,219]. Considering that microbial symbionts can influence the capacity of their coral hosts to acclimatize and adapt to environmental stressors, in this section we assess whether microbial dysbiosis can represent an adaptive mechanism, where the microbiota (i) senses changes in environmental conditions and acts as a buffer against these disturbances, (ii) senses physiological changes in the coral host before the first signs of deterioration, and favors recruitment of alternate, thermo-tolerant Symbiodiniaceae partners, or (iii) both of these occur (hypotheses are not mutually exclusive).

The adaptive dysbiosis hypothesis (ADH) stands on the principle that the microbiota confers resistance and resilience to its host via functional redundancy (reviewed in [220]), and is suspected to be involved in coral stress tolerance [221,222]. A large body of work has provided evidence for microbial flexibility (dynamic restructuring of microbial symbionts under changing environmental conditions) as a mechanism for coral environmental adaptation and bleaching tolerance (reviewed in [223]). Dysbiosis between corals and their associated microbiota may confer novel adaptive capabilities to environmental changes. Such an adaptive process occurs as long as the coral host can quickly revert back to a healthy state (eubiosis) by recovering a better suited Symbiodiniaceae assemblage, either by recruiting exogenous species from the surrounding environment (switching) or adjusting the relative abundance of native species from a low-abundance background (shuffling) [224,225]. The microbiota may act as a buffer to delay the coral–Symbiodiniaceae breakdown by maintaining essential functions when facing weak disturbances [167,173]. Microbial dysbiosis can thus provide a potential source of adaptive variation via diverse processes that provide physiological changes essential to support the coral–dinoflagellate symbiosis. Those processes include shifts in microbial gene expression [226,227], changes in microbiota taxonomic composition [130] and recruitment of new microbial species [186]. The restructuring of microbial communities, i.e., the addition or loss of microbial species, as well as changes in their relative abundances, is termed metagenomic plasticity [228], and has been suggested as an important mechanism of coral host plasticity and adaptation [31,130,161]. When exposed to strong disturbances that exceed the buffering capacity of the microbiota, corals can shift Symbiodiniaceae types to enhance resilience to environmental changes [225,229,230]. Both switching and shuffling processes can translate into transient bleaching states, waiting for the coral holobiont to adapt and recover to a healthy state. Elevated temperature and light incidence, along with the accumulation of ROS, photoinhibition, and metabolic dysfunctions of the cnidarian–Symbiodiniaceae symbiosis [231,232], may initiate the expulsion of the Symbiodiniaceae cells. If the environmental stressor persists and exceeds the buffering capacity of both microbial and dinoflagellate partners, then a “traumatic” dysbiosis occurs where coral bleaching is irreversible due to the permanent loss of the Symbiodiniaceae, often accompanied by the occurrence of opportunistic and invading pathogens. Accordingly, we propose the succession of three holobiont dysbiosis stages to define coral bleaching (Figure 3):(1)Adaptive dysbiosis without Symbiodiniaceae community restructuring;(2)Adaptive dysbiosis with Symbiodiniaceae community restructuring with or without transient bleaching;(3)Maladaptive/traumatic dysbiosis with irreversible loss of Symbiodiniaceae and invasion of opportunistic microbes leading to holobiont death.

### 3.1. Adaptive Dysbiosis without Symbiodiniaceae Community Restructuring

When corals are exposed to thermal stress (among others), major changes in microbiome composition occur, often associated with different bleaching states, but without inferring a causative link between the two [35,142]. Therefore, the genetic information encoded by the genomes of microorganisms can change instantly within a generation (metagenomic plasticity), contrary to the host genotype. Accordingly, under changing environmental conditions, metagenomic plasticity would potentially confer immediate adaptive capacity to the coral holobiont. Such a fast evolving capacity that microbiota confers to the holobiont results from various mechanisms, such as recruitment of new microbial species from the environment, horizontal gene transfers (plasmids, lysogenic phages, environmental DNA), and mutations, suggesting that coral microbiomes play a role not only in holobiont health, but also in its resilience.

Among these mechanisms, the increase in cyanobacteria and plastid ratio within microbial communities under stress has been identified as a potential mechanism to compensate for reduced photosynthetic activity linked to a lower abundance of Symbiodiniaceae cells [233], while a proliferation in nitrogen-fixing (diazotrophic) bacteria has been observed in corals exposed to elevated temperatures [175,234,235]. Although the exact mechanism behind the association between corals and diazotrophic bacteria remains to be solved, their presence is positively correlated with the abundance of dinoflagellates [166]. The loss of Symbiodiniaceae partners in bleached corals leads to a decrease in the holobiont’s capabilities to acquire nitrogen [236,237]. As such, diazotrophic bacteria can confer an additional source of nitrogen to thermally stressed corals and mitigate heat-induced coral bleaching [142,175].

The intake of new microbial symbionts is not always mandatory to maintain functional homeostasis. The microbiota can activate a natural defense mechanism against invading pathogens during coral dysbiosis. For instance, commensal bacteria are known to produce antibiotics and other metabolites involved in the enzyme inhibition (i.e., decrease in enzyme-related processes) [179,238]. Utilization of energetic reserves and/or heterotrophic feeding (i.e., zooplankton) represent another mechanism to maintain coral homeostasis during elevated temperatures [239,240]. Heterotrophic feeding can sustain coral energetic requirements during bleaching events, in which the supply of nutrients from the dinoflagellates decreases or ceases [239,240], and may contribute to coral thermal tolerance by providing an additional source of carbon [241]. The extent to which this process can meet coral nutritional needs during bleaching varies greatly across species, ranging from 0 to 35% in *Porites* species and up to 100% in *Montipora capitata* [240], while heterotrophic feeding is not enough to maintain coral homeostasis in the face of recurrent bleaching events [239].

Coral holobionts living in variable environments can express some level of metagenomic plasticity since they are forced to modify their physiological activity and/or their microbial composition frequently [130,242]. Compositional changes of the microbiota are often observed in response to stress, thus conferring short-term adaptation. If the new environmental conditions persist over time, then vertical transmission of these changes would allow a long-term adaptation. Such microbiota variations following a stress event have been demonstrated to be transmitted to the next generation [233]. The coral holobiont can therefore be considered as a selective unit.

### 3.2. Adaptive Dysbiosis with Symbiodiniaceae Community Restructuring with or without Transient Bleaching

First introduced by Buddemeier and Fautin [243], this holobiont dysbiosis stage involves both bacterial and dinoflagellate symbiont restructuring and stands on two assumptions: (i) that different Symbiodiniaceae species within a single genus can respond differently to environmental conditions [18,27] and (ii) that bleached corals can secondarily acquire new Symbiodiniaceae species directly from the environment [244]. Some experimental studies have demonstrated shifts in the taxonomic composition of coral-associated Symbiodiniaceae assemblages, where the intake of an alternative Symbiodiniaceae genus results in better suited physiological functions during temperature anomalies [225,229,230]. However, most coral-associated Symbiodiniaceae communities remain stable to retain photosynthetic function during acute thermal stress [245,246], or when recovering from stress events [247,248]. When recruitment of new dinoflagellate partners from the environment is needed, these alternative species are maintained in high abundance in the host’s tissue via shuffling or switching until the stressor is alleviated or stopped. Shuffled, and possibly switched, Symbiodiniaceae communities are heritable through maternal transmission, providing a fast-adaptive mechanism for corals exposed to changing environmental conditions [249]. Conversely, bleaching may also be triggered by the cnidarian–Symbiodiniaceae symbiosis shifting from a mutualistic to a parasitic relationship, when the availability of nutrients is reduced under thermal stress [128].

Because some *Durusdinium* species are considered thermo-tolerant, these better adapted Symbiodiniaceae can minimize coral bleaching and increase host survival under thermal stress [230,250,251,252], compared to species of other genera, such as *Cladocopium.* Although some *Durusdinium* species can potentially confer greater thermal tolerance, associating with this genus might represent a trade-off for corals, as for example they support slower coral growth compared to *Cladocopium* species [253,254]. Corals can also return to their original Symbiodiniaceae partnership while recovering from environmental stress (2–3 years after the bleaching event) [255], suggesting that associations with an alternative and more resistant species might only provide short-term benefits. Opportunistic dinoflagellate, such as *Durusdinium*, can replace other, less resistant species during bleaching to help the coral survive before returning to its beneficial association with its original partner. *Durusdinium* species might therefore act as transitional helpers rather than permanent partners.

The transition between two Symbiodiniaceae genera is costly in terms of energy, often resulting in a reduced number of symbionts for photosynthesis [256,257]. Because of their role in photosynthetic activity, the recruitment of cyanobacteria can potentially sustain important functions in the coral holobiont during this transitional period [22]. Such an increase in cyanobacteria during coral bleaching supports the partial contribution of the microbiota in maintaining the stability of the photosynthetic activity in order to meet the coral nutritional requirements during transitional bleaching (i.e., reversible loss of dinoflagellate symbionts). At last, latent viral infections of Symbiodiniaceae might be induced by light and/or heat stress [258], and infected Symbiodiniaceae are more sensitive to thermal stress [259]. Viral infections of Symbiodiniaceae have negative effects on thermal tolerance that most likely increase coral susceptibility to bleaching. In some cases, heterotrophic feeding may also provide the buffer time that is necessary for corals to switch or shuffle their associated dinoflagellate communities in response to increasing temperatures.

### 3.3. Madaptive/Traumatic Dysbiosis with loss of Symbiodiniaceae and Invasion of Opportunistic Microbes Leading to Holobiont Death

In cases of abrupt or prolonged elevated temperature, the microbiome can no longer maintain resistance against invading opportunistic pathogens (e.g., due to a decrease in the production of antimicrobial substances) and an irreversible breakdown in the holobiont equilibrium takes place, where the dinoflagellate symbiont loss is permanent [238]. Following coral mortality induced by thermal stress, shifts in reef ecosystems can be observed. Nitrogen production from dead corals has been documented to be as much as 30 times higher compared to healthy reefs [260], favoring an anoxic environment [261] and algae growth [260]. Endolithic boring communities dominated by *Ostreobium sp.* contribute to the dissolution of the calcium carbonate structure of dead corals, thus enhancing erosion by ocean acidification [262].

## 4. Microbiome Manipulations

Coral reefs are undergoing rapid decline globally as human activities have led to a gradual increase in sea temperature that surpasses the physiological tolerance of corals and their associated microorganisms. We are currently witnessing the most prolonged global coral die-off to date, mostly due to coral bleaching (from 2014 onwards) [263]. In 1998, 16% of global coral reefs perished [264], and 92% of the Great Barrier Reef showed some degree of bleaching in 2016 (1156 surveyed reefs) [265]. As we progress further into the Anthropocene, the fate of coral reefs under this unprecedented rate of environmental change is of particular concern and has driven the development of potential approaches for mitigating the effects of climate warming. Manipulations of rapidly evolving bacterial communities associated with the coral host might be one strategy to enhance coral tolerance to stress and bleaching. These new approaches include the isolation of natural microbiomes whose phenotypic traits are associated with increased survival and resilience (probiotics) [266], and the genetic modification of coral-associated microorganisms (microbiome engineering) to improve host performance and fitness in the face of climate change [98]. Although microbiome manipulations may represent a key strategy to improve coral phenotype and ecosystem functioning, these efforts are still challenging. Challenges include the lack of knowledge on the mechanisms causing bleaching, the role of microbes in the development of bleaching, the identity of stable and crucial symbionts promoting coral health, and the processes involved in the establishment and maintenance of the coral-microbiome symbiosis, in addition to the difficulty of isolating microorganisms [142]. Nevertheless, recent manipulations of bacteria associated to the coral *P. damicornis* have proven to be useful in lessening the effects of bleaching through the addition of a consortium of native putatively beneficial microorganisms for corals (pBMCs), including five *Pseudoalteromonas* sp., a *Halomonas taeanensis* and a *Cobetia marina*-related species strains [58]. Such microbial inoculations were also successful in increasing the resistance of corals to oil pollution, as well as promoting the degradation of water-soluble oil fractions [267]. Because coral thermal tolerance is partly dependent on the Symbiodiniaceae species hosted, inoculations of heat-adapted photosymbionts through experimental evolution can represent another strategy of improving coral health and bleaching tolerance [268,269].

Despite the potential for harnessing the benefits of coral microbiomes, their manipulations are not enough to overcome the impact of human-induced climate change and other threats (e.g., pollution, eutrophication, overfishing) on coral reef ecosystems. Other local/small scale actions must be taken, in combination with climate action, to protect and restore these diverse and productive ecosystems. Ecological restoration, creation of marine protected areas, strict monitoring of physicochemical parameters, and international collaboration sharing data on corals are only a few examples that would help in mitigating the current and unprecedented coral decline [270,271,272]. The establishment of a holobiont health database is therefore critical to obtain a more integrated understanding of factors that are threatening coral ecosystems. Ongoing efforts to conserve coral reefs should thus rely on field and laboratory experiments to generate robust multi-omics data. The application of single “omics”-based approaches—which refer to technologies used to explore the roles, relationships, and functions of a large family of molecules that make up the cells of an organism, such as genomics (genes), transcriptomics (mRNAs), proteomics (proteins), and epigenomics (epigenetic marks)—has helped to identify molecular signatures linked to the plasticity and adaptation of the coral animal and its associated microbes [82,83,84,273,274,275]. Nevertheless, more investigations that employ multi-omics approaches to the coral holobiont are required to facilitate the study of associations and interactions within and across these omics layers.

## Figures and Tables

**Figure 1 microorganisms-08-01682-f001:**
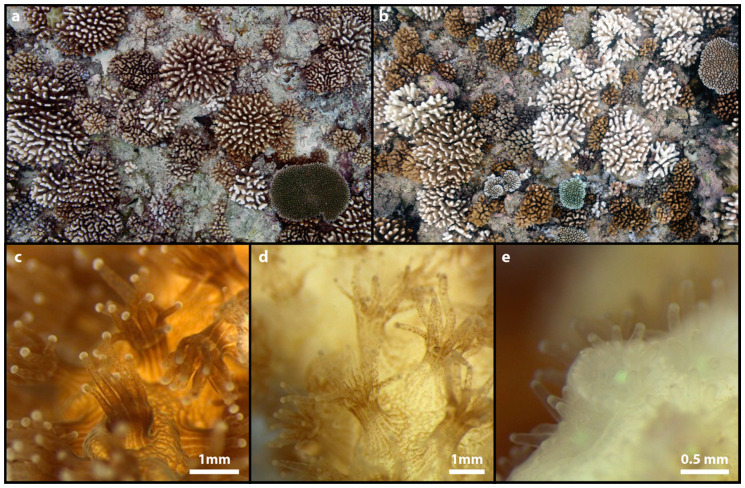
Coral bleaching. The upper photographs show a reef landscape (reef slope) before (**a**) and during (**b**) the massive bleaching event of 2016 at Moorea, French Polynesia. The bottom macro-photographs show a close-up lateral view of the polyps of the scleractinian coral *Pocillopora damicornis* experiencing bleaching: unbleached (healthy) polyps with visible Symbiodiniaceae (**c**), partially bleached polyps with low density of Symbiodiniaceae (**d**), and bleached polyps without Symbiodiniaceae and transparent tissue (**e**). Photographs are courtesy of Y. Chancerelle (**a**, **b**), and L. Hédouin (**c**–**e**) (CRIOBE, Moorea).

**Figure 2 microorganisms-08-01682-f002:**
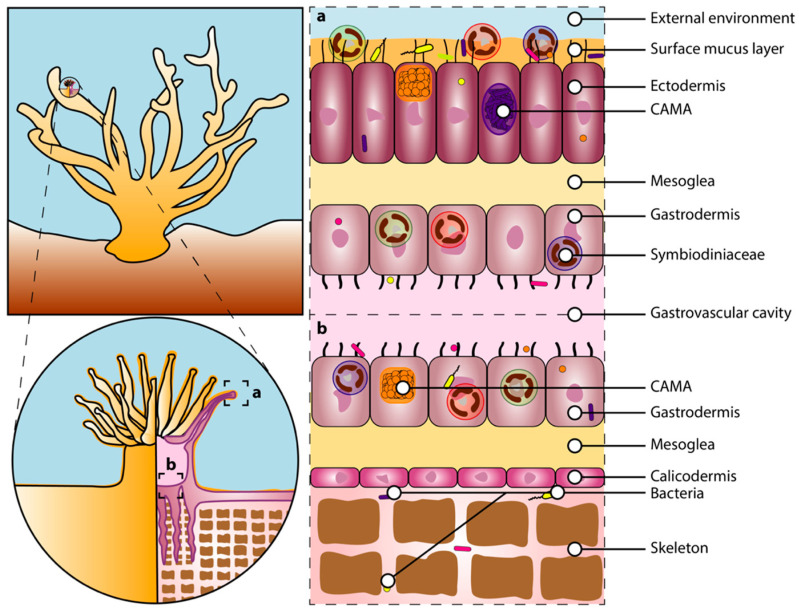
Coral microhabitats and their associated microorganisms. The diagram on the left illustrates a healthy coral (**top**) and a close-up transversal view of one polyp to illustrate the tissue layers (**bottom**). Diagrams on the right represent a close up of a cross-section of the coral tissues of a tentacle (**a**) and the basal part of the polyp (**b**), showing the distribution of the Symbiodiniaceae, bacteria, and cell-associated microbial aggregates (CAMAs) in the external environment, surface layer, ectodermis, gastrodermis, gastrovascular cavity, and skeleton. The size of the Symbiodiniaceae and microbes has been modified for illustration purposes. For an accurate size representation, see [90,91,92].

**Figure 3 microorganisms-08-01682-f003:**
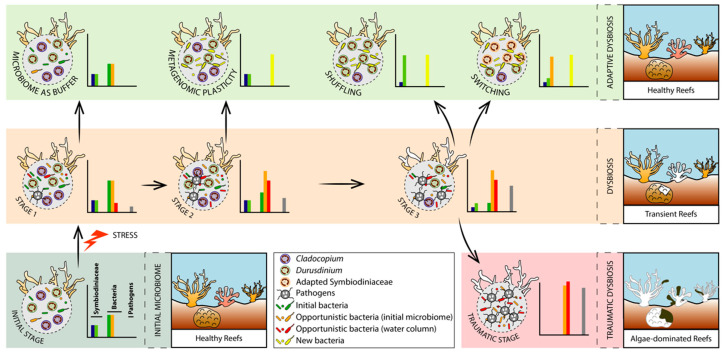
Holobiont dysbiosis phases associated with their coral bleaching states. The diagram illustrates an environmental stressor (increased temperature) inducing the disruption of the initial microbiome, leading to a dysbiosis state (stage 1) where the coral holobiont remains healthy (no sign of bleaching) despite the increase in the abundance of native opportunistic bacterial species and the settlement of invading pathogens and other opportunistic bacteria from the surrounding environment. The microbiota could act as a buffer and revert to its initial bacterial composition, representing an adaptive dysbiosis; or the environmental stressor could exceed the buffer capacity of the microbiota leading to another dysbiosis state (stage 2), where opportunistic bacteria and pathogens will slowly replace the initial microbiota, while the coral holobiont still remains healthy. Metagenomic plasticity, here referring to the addition of novel bacterial species and functions (but changes in relative abundances of native bacteria can also occur), either leads to another adaptive dysbiosis; or the environmental stressor exceeds once again the buffer capacity of the microbiota leading to another dysbiosis state (stage 3), where transient bleaching can be observed. Corals can revert to a healthy state by recovering a better suited Symbiodiniaceae assemblage, either by adjusting the relative abundance of native species from a low-background (shuffling), here referring to the increase in heat-tolerant *Durusdinium* species; or by recruiting exogenous species from the surrounding environment that are better adapted to elevated temperature (switching). If the stressor causes an abrupt change in the reef environment or persists for long periods, the adaptive dysbiosis will not act as a mechanism of environmental adaptation. In that case, the traumatic dysbiosis occurs with the permanent loss of the Symbiodiniaceae partners leading to coral death, and ultimately to a reef ecosystem dominated by algae and cyanobacteria (phase shifts). Each dotted circle represents the microbiome of a coral at each of the dysbiosis stages described, while the bar plots illustrate the declining number of “good microorganisms” and the increasing number of “bad” ones within the coral microbiome at each of the dysbiosis stages. The number and size of the Symbiodiniaceae and microbes within the coral tissues has been modified for illustration purposes.

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
