# Peer review of "Defining Coral Bleaching as a Microbial Dysbiosis within the Coral Holobiont"

_microorganisms, 2020, doi:10.3390/microorganisms8111682_

Round 1
Reviewer 1 Report
Review of Boilard et al. Microorganisms
Summary
In this article, the authors have nicely reviewed progress in the study of coral bleaching. Unlike many other bleaching reviews, which tend to focus ONLY on the coral hosts and their dinoflagellate endosymbionts, the authors instead make a compelling case for considering three (or more) compartments: cnidarians, dinoflagellates, and other microbes (both eukaryotic and prokaryotic). I am especially impressed by the breadth given that only one author has published on corals before. Despite several key references nevertheless having been omitted (pointed out below), the article is exceptionally well researched, and I like the eubiosis to dysbiosis story. We recently found that anemone-Symbiodiniaceae endosymbiosis go from being mutualistic to parasitic (the latter being the parasite) in starved anemones (Peng et al. PeerJ in press probably around Oct. 25th), and I know others have seen such shifts. It does make me wonder if microbial shifts occur BEFORE bleaching, DURING bleaching, or as a RESULT of bleaching? I know bacterial infections are not a major cause of bleaching, but maybe the initial stages of stress could cause a microbiome shift that then exacerbates the bleaching process, creating a sort of snowball effect. Is it possible to state this based on the collective work that has been done? It’s okay if not, but if this is known, that could make for a nice paragraph to include. I think if the authors are able to improve the figures (and potentially add more) and address the comments listed below, this will be a nice contribution to the field. If felt needed by the editorial board, I would be happy to double-check it once such revisions have been made.
Major comments
- I would recommend including 1-2 additional figures, if for no other reason than to be commensurate with the length and scope of the article (both being impressive). Maybe you could have some coral reef images or some bleached coral photos. Or maybe a picture of a diseased coral, as well. Maybe you could create a Venn diagram of the three compartments, though I feel like this may have already been done. There is one in this thesis, but it’s not very good: http://eprints.uni-kiel.de/47235/1/Master%20Thesis%20Talisa%20D%C3%B6ring.pdf. Maybe you could make a nicer, more thorough version?
- I like the message of Figure 2 but not its presentation. See my comments below.
- There are numerous works by Mayfield et al. from Taiwan where they actually tried to estimate the biological makeup of the coral holobiont using genetic informatic (namely RNAs, DNAs, and proteins). They then use these data to show how coral holobionts are spatio-temporally dynamic, even when bleaching is not occurring. I think these works will be of interest to the authors and could be cited since they are integral to the understanding of the topic at hand.
- Lines 126-138. I would scale back on this section since many of the findings were recently debunked. Basically, there is no correlation between gene expression and protein concentrations in reef corals or their endosymbionts, so we’ve had to scale back our interpretation of these “front-loading” and immunology data (since they were, unfortunately, based on gene expression alone). In the future, we need to instead focus on the PROTEINS encoded by these genes since they actually carryout the work in the cells (mRNA data mean nothing unless the proteins follow suit). Same goes for other sections (e.g., 375-376) unless the data are backed up by protein-based findings. Or maybe you can just keep the text the same but add a cautionary statement: “Keep in mind that much of what we know about coral molecular biology comes from gene expression data alone; however, a number of papers (Mayfield et al. 2014, 2016, 2018, etc.) have shown an absence or correlation or congruency between gene expression and concentration of the encoded protein, meaning that it is premature to use mRNA concentration data alone to make inferences into the biology of coral or dinoflagellate cells.” These same works are multi-omics analyses, which you advocate. Maybe you could tag them onto the end of the work (where you do so), so as not to have to change and reformat all the references!
- Not sure if you mentioned it anywhere, but if not I would recommend it: corals, being the holobionts that they are, can both adapt AND acclimatize within a single individual. A coral that has undergood symbiont switching has evolved at the hologenome level. The cnidarian genome remains unchanged, but by switching in and out various microbes, the holobiont is in a constant state of evolution (literally). At the same time, the individual compartments might also be acclimatizing to environmental change, adding yet another layer of complexity to the story! Lines 231-233, you somewhat mention this, but it’s even simpler: it doesn’t “drive future evolution;” it is “evolution in action.” A coral holobiont is almost surely to be evolving continually (though whether such evolution confers a benefit is another story since the microbiome could change without benefiting the coral holobiont, i.e., commensalism).
- No discussion of bleaching is complete without some attention given to the role of/importance of light. Corals basically won’t bleach in a black box. I would provide a few sentences about this somewhere. I wonder what the influence of light is on the bacteria? On the one hand, you would think it would more significantly affect the phototropic compartments, but I guess light could indirectly influence all members of the holobiont. Doesn’t merit a whole new section, maybe just pepper a few sentences about light and holobiont health/homeostasis at a few points.
- We have a new article out that show that symbionts become parasitic when their hosts are starved. While this was done in sea anemones, it might be of interest to the authors: Peng et al. PeerJ. I think it will be available online soon. You do mention shifts to parasitism in the bacteria, but you may want to mention somewhere that the “friendly” dinoflagellates actually can become parasitic, as well (there are other works on this besides ours). Simon Davy likely also has some parasitism/dysbiosis work recently.
- There is a “chicken and the egg” situation that I have often wondered about, and I wonder if there is now enough literature and information out there to address it. Based on your research for this review, do you think coral bleaching LEADS to shifts in the non-Symbiodiniaceae microbial communities, coincides with such shifts, or is caused by such shifts? I would argue against the latter, but who knows; maybe a microbial shift occurs before the ROS are produced, for instance. Has anyone done a study that could answer this?
- Although the manuscript is very well written, I would nevertheless recommend that it be proofread by a native English speaker. My list of typo and grammatical corrections below is non-exhaustive.
Minor comments
Overall
- It is a semantic issue, but dinoflagellates are technically microbes. At the avoidance of having to say “non-Symbiodiniaceae microbes” or something equally cumbersome, maybe you can directly state that “all non-Symbiodiniaceae microbes, including other eukaryotic algae, fungi, bacteria, viruses, etc.” will hereafter be termed ‘microbes’.”
- Similarly, only use “dinoflagellate” or even “Symbiodiniaceae” when talking about the intra-gastrodermal mutualists. “Microalgae” or “algae” could refer to non-Symbiodiniaceae algae (unless you ARE wanting to include all associated algae). My guess is that, though, in most cases you equate Symbiodiniaceae to algae (and vice versa).
- Keywords should be alphabetized.
- In most cases, taxonomic terms need not be capitalized, only genus and family names. Examples: dinoflagellates, ascomycete, basidiomycetes, corallicolids, etc.
- Numbers lower than 10 should be written out (e.g., “four” instead of “4”).
Abstract
- The abstract is well-written and informative, but I think there is too much common knowledge. I would cut it in half. “Coral reefs are threatened by climate change, in particular the rising temperatures, which elicit bleaching. This review provides an overview….”
- Line 32: no need for the last two terms. “death” is the word you are looking for. You can define those other terms later.
Main text
- Lines 40-42. This sentence should be reworded since it reads as if “coastal protection” and “medicines” are jobs.
- Line 41: “that” è “and”
- Line 43: “constitute” è “are” (constitute implies being made from)
- Line 43: “foundation”è”foundational”
- Line 46: cnidarian host
- Line 46: family Symbiodiniaceae
- Line47: this is a run-on sentence that should be broken into two. Also, see my comment above about mentioning whether or not you include Symbiodiniaceae in the catch-all of “microbiota” (or “microbiome”) in the remainder of the article.
- Line 49 and elsewhere: At this point, I would only use “holobiont” and not “meta-organism.”
- Lines 50-51: remove “continuous” since it is not always the case.
- Line 51: Symbiodiniaceae partners è “dinoflagellate endosymbionts”
- Lines 54-56: This is technically not true as stated since it implies a transfer of DNA within the holobiont. Instead, you should simply state that the hologenome, as well as the interactions among the various compartments, drive the biology of the composite holobiont.
- Line 58: host-microbe interactionèholobiont
- Line 58: I wouldn’t say “easily.” Instead, it could be “readily” or “oftentimes.” Better phrasing could be: “Holobionts can transition from representing eubioses to dysbiosis as, for instance, environmental conditions deteriorate.”
- Line 62: some of which directly or indirectly impact holobiont homeostasis
- 64: areèis, haveè In the same sentence, “several” è “numerous”
- Line 67: remove “pulse” since you are also referring to longer-term events.
- Line 71: dysbiotic
- Line 72: This is no an exhaustive list, since you only mention 2-3 of the 5-6 different mechanisms (see Gates et al. 1992). You just need to emphasize in parentheses that these are just a few of the mechanisms. Or, alternatively, you could list all of them (e.g., host cell detachment).
- Lines 74-76: This has recently been called into question. See Nielsen et al. The ROS actually don’t end up in the host coral cells! I think you should mention that the mechanism is still far from being fully understood.
- Line 76: “Since the 1980s…” should start a new paragraph.
- Line 87: “host’s”è “host”
- Lines 194-196: This is not true. Cladocopium is found alllll over the place: high light, low light, shallow, deep, numerous species, etc. Check out Mayfield et al. (2017, PLoS ONE) for a spatio-temporal analysis of Cladocopium presence in the South Pacific.
- Not all Durusdinium are heat-tolerant. The section around line ~550 should be worded more cautiously (e.g., “Some Durusdinium sp….”).
- Line 254: I know it’s an illustration, but the CAMAs seen in actual corals are much smaller than those shown in the figure. I wonder if this should be mentioned in the caption. There are TONS of microbes in coral, but the actual amount of SPACE they occupy (e.g., biomass and volume) is small. Check out some of our SEM and TEM images (Peng et al., 2011, Proteomics, Mayfield et al. 2013, Coral Reefs, works of Chii-Shiarng Chen).
- Line 285: Orbicella faveolata, not Montastrea (which is the old name).
- Line 328: No need to abbreviate quorum sensing since you never mention it again.
- Lines 343-346: Exactly. This applies to ALL the microbial+bleaching work to date. There are a few phrases elsewhere that imply that viruses or bacteria directly cause bleaching, which, to my knowledge, has long been debunked (e.g., lines 395-396 and reference195, which did not find virus-induced bleaching).
- Line 384 and elsewhere: I think microbiota should almost universally be “microbiome” in the article.
- In the same paragraph, “dinoflagellates” does not need to be capitalized, and there are many English mistakes. I would have this section revised by a native English speaker.
- Line 398: Despite (never plural)
- Line 425: “correlate” is the wrong word. You mean “are associated with” or “have been documented alongside.” This is sort of a chicken and the egg situation; does the stress event lead to a microbial shift that then exacerbates bleaching, or does the bleaching process disrupt the coral-dinoflagellate endosymbiosis?
- Line 431: love this idea, and I know Vega-Thurber is trying it. I will, too. Do these microbiome shifts occur BEFORE the downstream effects we can see or IN CONCERT WITH them? Hopefully, it’s the former (or else the diagnostic capacity falls to 0).
- Around lines 450: I like the idea that redundancy could reduce potential bleaching impacts, but in this section are you talking about background dinoflagellate symbiont types or bacteria? It could be both, so just make sure to specify which microbes you are referring to. See my suggestion elsewhere about changing the wording to avoid confusion (since Symbiodiniaceae are technically microbes, as well).
Figures
- For an article of this length, I would like to see an additional 1-2 figures, even if it is something generic like a before and after picture of a bleached coral. Of course, don’t force it if there is really nothing else that could benefit the article.
- Figure 2. I like the idea of the figure more than I like the visuals. One issue is the scale. The dinoflagellates occupy 90% or more of the cell volume, yet I worry that readers will think of these circles as cells, in which case the scale is way off. Maybe you could use the polyps drawn elsewhere to show this idea instead. BTW, it should be “shuffling.” The icons in the circles are also too small to read, to where the reader cannot glean the major message you are trying to make. I wonder if, below select circles, you could have a very simple plot that shows the declining number of “good microbes” and the increasing number of “bad” ones, as you go from stage to stage? One line would be ever-increasing (the bad) and another line (colored to match the good microbes) would be ever-decreasing. Again, I like this idea, and I don’t disagree with anything in the figure. I just wonder if you can make it more visually compelling and easier to interpret?
- Figure 2: Should it be “acclimatized” instead of “adapted” Symbiodiniaceae? An individual cell can, of course, not adapt, but if you mean a cell from a population of symbionts that adapted to life at high temperatures, then “adapted” could still be the right term. But if you mean a symbiont that became stronger on account of, for instance, prior stress exposure, then “acclimatized” would be the correct term. See my treatise elsewhere on the notion that corals can simultaneously evolve and acclimatize. Not sure if this could somehow be incorporated into the figure without making it too convoluted (especially given my prior suggestions), but just a thought.
References
- The references are inconsistently formatted. Sometimes the article’s title is capitalized in its entirety and sometimes not. I think for Microorganisms, only the first word is capitalized.
- Species names should be italicized in the references.

Author Response
Responses to the Reviewer 1 comments:
In this article, the authors have nicely reviewed progress in the study of coral bleaching. Unlike many other bleaching reviews, which tend to focus ONLY on the coral hosts and their dinoflagellate endosymbionts, the authors instead make a compelling case for considering three (or more) compartments: cnidarians, dinoflagellates, and other microbes (both eukaryotic and prokaryotic). I am especially impressed by the breadth given that only one author has published on corals before. Despite several key references nevertheless having been omitted (pointed out below), the article is exceptionally well researched, and I like the eubiosis to dysbiosis story. We recently found that anemone-Symbiodiniaceae endosymbiosis go from being mutualistic to parasitic (the latter being the parasite) in starved anemones (Peng et al. PeerJ in press probably around Oct. 25th), and I know others have seen such shifts. It does make me wonder if microbial shifts occur BEFORE bleaching, DURING bleaching, or as a RESULT of bleaching? I know bacterial infections are not a major cause of bleaching, but maybe the initial stages of stress could cause a microbiome shift that then exacerbates the bleaching process, creating a sort of snowball effect. Is it possible to state this based on the collective work that has been done? It’s okay if not, but if this is known, that could make for a nice paragraph to include. I think if the authors are able to improve the figures (and potentially add more) and address the comments listed below, this will be a nice contribution to the field. If felt needed by the editorial board, I would be happy to double-check it once such revisions have been made.
Response: We sincerely appreciate all the comments and suggestions that the reviewer brought and are very thankful for his/her thorough review. Also, we highly value that the reviewer applauded “our vision” of including many holobiont components in our review to define coral bleaching as a microbial dysbiosis. We also included all the additional references suggested by the reviewer and modified the figures as requested.
Regarding the current knowledge about microbial shifts occurring before, during, or as a result of bleaching, we were not able to find any studies with compelling observations to disentangle this debate. However, our personal opinion leans towards the first hypothesis, i.e. microbial communities may start to change before the bleaching occurs.
Major comments
(1) I would recommend including 1-2 additional figures, if for no other reason than to be commensurate with the length and scope of the article (both being impressive). Maybe you could have some coral reef images or some bleached coral photos. Or maybe a picture of a diseased coral, as well. Maybe you could create a Venn diagram of the three compartments, though I feel like this may have already been done. There is one in this thesis, but it’s not very good: http://eprints.uni-kiel.de/47235/1/Master%20Thesis%20Talisa%20D%C3%B6ring.pdf. Maybe you could make a nicer, more thorough version?
Response: We included an additional figure showing photographs of a reef landscape at Moorea, French Polynesia, before and during the massive bleaching event of 2016, and a close-up of the polyps of the coral Pocillopora damicornis experiencing bleaching, i.e. healthy, partially bleached, and bleached. Please see the attachment.
(2) I like the message of Figure 2 but not its presentation. See my comments below.
Response: We modified the figure according to the reviewer’ suggestions and we incorporated diagrams for each dysbiosis state to visualize the gain and loss of the different symbiotic partners. Please see the detailed response below (in answer to the figure item (2)).
(3) There are numerous works by Mayfield et al. from Taiwan where they actually tried to estimate the biological makeup of the coral holobiont using genetic information (namely RNAs, DNAs, and proteins). They then use these data to show how coral holobionts are spatio-temporally dynamic, even when bleaching is not occurring. I think these works will be of interest to the authors and could be cited since they are integral to the understanding of the topic at hand.
Response: We added numerous works by Mayfield et al. from Taiwan in the revised manuscript. Among them are Mayfield et al. 2013; 2014; 2016; 2018. Thank you for sharing these studies.
(4) Lines 126-138. I would scale back on this section since many of the findings were recently debunked. Basically, there is no correlation between gene expression and protein concentrations in reef corals or their endosymbionts, so we’ve had to scale back our interpretation of these “front-loading” and immunology data (since they were, unfortunately, based on gene expression alone). In the future, we need to instead focus on the PROTEINS encoded by these genes since they actually carry out the work in the cells (mRNA data mean nothing unless the proteins follow suit). Same goes for other sections (e.g., 375-376) unless the data are backed up by protein-based findings. Or maybe you can just keep the text the same but add a cautionary statement: “Keep in mind that much of what we know about coral molecular biology comes from gene expression data alone; however, a number of papers (Mayfield et al. 2014, 2016, 2018, etc.) have shown an absence or correlation or congruency between gene expression and concentration of the encoded protein, meaning that it is premature to use mRNA concentration data alone to make inferences into the biology of coral or dinoflagellate cells.” These same works are multi-omics analyses, which you advocate. Maybe you could tag them onto the end of the work (where you do so), so as not to have to change and reformat all the references!
Response: We agree with the reviewer. Accordingly, we added this cautionary statement at new lines 199-201: “Importantly, gene expression has been found uncorrelated or incongruent with the concentration of the encoded proteins [Mayfield et al. 2014, 2016, 2018], pointing out conclusions drawn from the exclusive use of mRNA concentration data should be cautiously interpreted.” We also modified the text at new lines 491-492 to incorporate studies on the absence of congruency between gene expression and concentration of the encoded proteins: “Rising temperature can also increase virulence by expressing genes causing dysbiosis (but see [Mayfield et al. 2014, 2016, 2018] for incongruence between gene expression and encoded proteins).” At last, we incorporated Mayfield et al. 2014, 2016, 2018 at new lines 803-827, where we mention the importance of multi-omics analyses.
(5) Not sure if you mentioned it anywhere, but if not I would recommend it: corals, being the holobionts that they are, can both adapt AND acclimatize within a single individual. A coral that has undergood symbiont switching has evolved at the hologenome level. The cnidarian genome remains unchanged, but by switching in and out various microbes, the holobiont is in a constant state of evolution (literally). At the same time, the individual compartments might also be acclimatizing to environmental change, adding yet another layer of complexity to the story! Lines 231-233, you somewhat mention this, but it’s even simpler: it doesn’t “drive future evolution;” it is “evolution in action.” A coral holobiont is almost surely to be evolving continually (though whether such evolution confers a benefit is another story since the microbiome could change without benefiting the coral holobiont, i.e., commensalism).
Response: We modified the text at new lines 315-318 to simplify our thoughts on the manner: “Consequently, changes in terms of the microbial taxonomic composition and functional repertoires (i.e. the metagenome) represent an evolutionary process in action that could potentially produce heritable adaptive phenotypic traits [125]. Although the cnidarian genome remains unchanged, the coral holobiont is continuously evolving.
(6) No discussion of bleaching is complete without some attention given to the role of/importance of light. Corals basically won’t bleach in a black box. I would provide a few sentences about this somewhere. I wonder what the influence of light is on the bacteria? On the one hand, you would think it would more significantly affect the phototropic compartments, but I guess light could indirectly influence all members of the holobiont. Doesn’t merit a whole new section, maybe just pepper a few sentences about light and holobiont health/homeostasis at a few points.
Response: Unfortunately, we are not aware of any studies so far describing the effects of light on bacteria, but we added this information for the cnidarian-Symbiodiniaceae symbiosis at new lines 606-608: “Elevated temperature and light incidence, along with the accumulation of ROS, photoinhibition, and metabolic dysfunctions of the cnidarian-Symbiodiniaceae symbiosis [Smith et al. 2005, Gardner et al. 2017], may initiate the expulsion of the Symbiodiniaceae cells.”
(7) We have a new article out that show that symbionts become parasitic when their hosts are starved. While this was done in sea anemones, it might be of interest to the authors: Peng et al. PeerJ. I think it will be available online soon. You do mention shifts to parasitism in the bacteria, but you may want to mention somewhere that the “friendly” dinoflagellates actually can become parasitic, as well (there are other works on this besides ours). Simon Davy likely also has some parasitism/dysbiosis work recently.
Response: We added this information at new lines 713-715: “Conversely, bleaching may also be triggered by the cnidarian-Symbiodiniaceae symbiosis shifting from a mutualistic to a parasitic relationship, when the availability of nutrients is reduced under thermal stress [Morris et al. 2019].”
(8) There is a “chicken and the egg” situation that I have often wondered about, and I wonder if there is now enough literature and information out there to address it. Based on your research for this review, do you think coral bleaching LEADS to shifts in the non-Symbiodiniaceae microbial communities, coincides with such shifts, or is caused by such shifts? I would argue against the latter, but who knows; maybe a microbial shift occurs before the ROS are produced, for instance. Has anyone done a study that could answer this?
Response: It is indeed the “chicken and the egg”, and to our knowledge, there is no study that monitors non-Symbiodiaceae microbial communities right before the ROS are produced. This is probably because these changes occur in a short time (i.e. minutes to hours) and space scales (i.e. microns). The exact bleaching mechanism is still unknown, and the microbial communities' role is still poorly understood. We would also argue against the latter (microbial shifts cause bleaching). However, the microbial communities may start to change before the bleaching occurs because the coral microhabitats also change before the bleaching. For example, the mucus tissue layer, epithelium, and gastrodermis change in their thickness. Ainsworth et al. (2008) reported tissue thickness changes 3C before the bleaching threshold. For personal observation, mucus production increases hours before colonies begin to be pale (this observation is also species-specific). These are not only physical changes but most likely represent also a nutritive change for microbes. No doubt, this is an interesting question, and we are glad the reviewer offered thoughts about it.
(9) Although the manuscript is very well written, I would nevertheless recommend that it be proofread by a native English speaker. My list of typo and grammatical corrections below is non-exhaustive.
Response: As suggested by the reviewer, the manuscript has been proofread by a native English speaker. Also, we revised the list of typo and grammatical errors.
Minor comments
R1.1: It is a semantic issue, but dinoflagellates are technically microbes. At the avoidance of having to say “non-Symbiodiniaceae microbes” or something equally cumbersome, maybe you can directly state that “all non-Symbiodiniaceae microbes, including other eukaryotic algae, fungi, bacteria, viruses, etc.” will hereafter be termed ‘microbes’.”
Response: We agree with the reviewer and revised the manuscript accordingly. We are now referring to the non-Symbiodiniaceae microorganisms as microbes, including other eukaryotic algae, fungi, bacteria, viruses, and others. In this current manuscript, we also refer to the microbiome as the combination of both the Symbiodiniaceae and other microbes. As such, we had decided to make a distinction between the microbiome (Symbiodiniaceae and non-Symbiodiniaceae microbes) and microbiota (only non-Symbiodiniaceae microbes).
R1.2: Similarly, only use “dinoflagellate” or even “Symbiodiniaceae” when talking about the intra-gastrodermal mutualists. “Microalgae” or “algae” could refer to non-Symbiodiniaceae algae (unless you ARE wanting to include all associated algae). My guess is that, though, in most cases you equate Symbiodiniaceae to algae (and vice versa).
Response: Thank you for noticing this error. We revised the manuscript in order to only refer to dinoflagellate and Symbiodiniaceae for the intra-gastrodermal symbionts.
R1.3: Keywords should be alphabetized.
Response: We alphabetized the keywords as suggested.
R1.4: In most cases, taxonomic terms need not be capitalized, only genus and family names. Examples: dinoflagellates, ascomycete, basidiomycetes, corallicolids, etc.
Response: Based on The International Code of Nomenclature of Prokaryotes (International Code of Nomenclature of Prokaryotes), rule 7 entitled Names of Taxa above the Rank of Genus up to and including Order, and rule 8 entitled Names of Taxa above the Rank of Order which state that the name of a taxon above the rank of the genus should be written with an initial capital letter, we have used capitals in the cases that refer to a taxonomic hierarchy and lower case for common names.
R1.5: Numbers lower than 10 should be written out (e.g., “four” instead of “4”).
Response: We revised the manuscript accordingly.
R1.6: The abstract is well-written and informative, but I think there is too much common knowledge. I would cut it in half. “Coral reefs are threatened by climate change, in particular the rising temperatures, which elicit bleaching. This review provides an overview….”
Response: In order to reduce the abstract as suggested, we remove this part: “Coral reefs are one of the most productive and diverse ecosystems on Earth. However, climate warming is occurring at an unprecedented rate and has negatively affected coral reefs worldwide. Corals are the foundation of reef ecosystems, yet they are relatively poor at surviving rapid environmental changes. Corals are meta-organisms composed of the cnidarian host animal, symbiotic dinoflagellates, and a suite of diverse microorganisms, collectively termed the coral holobiont.”
R1.7: Line 32: no need for the last two terms. “death” is the word you are looking for. You can define those other terms later.
Response: Maladaptive/traumatic dysbiosis were replaced by death at new line 27 as suggested. Nevertheless, this term was kept to define the succession of the three holobiont stages related to coral bleaching.
R1.8: Lines 40-42. This sentence should be reworded since it reads as if “coastal protection” and “medicines” are jobs.
Response: We modified the text at new lines 36-38 to: “This ecosystem also delivers key services such as fisheries [9,10], tourism-based industries [11], coastal protection [12], and medicines [13-15], sustaining the welfare and livelihoods of millions of people.”
Line 47: this is a run-on sentence that should be broken into two. Also, see my comment above about mentioning whether or not you include Symbiodiniaceae in the catch-all of “microbiota” (or “microbiome”) in the remainder of the article.
Response: We modified the text at new lines 40-67 to: “Corals are meta-organisms and are formed by a dynamic multipartite relationship between the cnidarian host, its endosymbiotic dinoflagellate algae (family Symbiodiniaceae [18]), and a suite of other non-Symbiodiniaceae microbes [22], hereafter termed the microbiota or microbes. The microbiota includes prokaryotes (archaea and bacteria) [19], eukaryotes (fungi and non-Symbiodiniaceae protists), and viruses [20,21]. In the present review, the dinoflagellate algae and microbiota are collectively termed the coral microbiome.”
Line 49 and elsewhere: At this point, I would only use “holobiont” and not “meta-organism.”
Response: We modified the manuscript accordingly from new line 76.
Lines 54-56: This is technically not true as stated since it implies a transfer of DNA within the holobiont. Instead, you should simply state that the hologenome, as well as the interactions among the various compartments, drive the biology of the composite holobiont.
Response: We modified the text at new lines 73-75 to: “The hologenome and the host-Symbiodiniaceae-microbiota interaction drive the biology of the holobiont [21,33] and ultimately define its phenotype [31].”
Line 58: I wouldn’t say “easily.” Instead, it could be “readily” or “oftentimes.” Better phrasing could be: “Holobionts can transition from representing eubioses to dysbiosis as, for instance, environmental conditions deteriorate.”
Response: We rephrased as suggested at new lines 76-78: “The stability of the holobiont is fragile. Holobionts can transition from eubiosis (healthy state of the holobiont) to dysbiosis (unhealthy, disrupted state of the holobiont) as, for instance, environmental conditions deteriorate.”
Line 62: some of which directly or indirectly impact holobiont homeostasis
Response: We modified accordingly.
Line 72: This is no an exhaustive list, since you only mention 2-3 of the 5-6 different mechanisms (see Gates et al. 1992). You just need to emphasize in parentheses that these are just a few of the mechanisms. Or, alternatively, you could list all of them (e.g., host cell detachment).
Response: We added few other described mechanisms as suggested at new line 92: “exocytosis, detachment or necrosis pathways [Gates et al.1992]”.
Lines 74-76: This has recently been called into question. See Nielsen et al. The ROS actually don’t end up in the host coral cells! I think you should mention that the mechanism is still far from being fully understood.
Response: We modified the text at new lines 94-96 to incorporate Nielsen et al. 2018: “Although evidence suggests that the production of reactive oxygen species (ROS) in the Symbiodiniaceae cells is the most likely cause of their expulsion [39], the bleaching mechanism is still far from being fully understood [Nielsen et al. 2018].”
Line 76: “Since the 1980s…” should start a new paragraph.
Response: “Since the 1980s now starts a new paragraph as suggested.
Lines 194-196: This is not true. Cladocopium is found alllll over the place: high light, low light, shallow, deep, numerous species, etc. Check out Mayfield et al. (2017, PLoS ONE) for a spatio-temporal analysis of Cladocopium presence in the South Pacific.
Response: We modified the text to specify that Cladocopium are considered depth-generalists at new lines 278-280: “On the other hand, Cladocopium species are considered depth-generalists, although populations of shallow habitats can exhibit greater assemblage diversity compared to deeper habitats at some reef locations [Eckert et al. 2020].”
Not all Durusdinium are heat-tolerant. The section around line ~550 should be worded more cautiously (e.g., “Some Durusdinium sp….”).
Response: We modified the text at new line 277 to: “with some Durusdinium species being the most thermal tolerant”, and similarly at new lines 716-718: “Because some Durusdinium species are considered thermo-tolerant, these better adapted Symbiodiniaceae species can minimize coral bleaching and increase host survival under thermal stress [226,245-247], compared to others species of other genera, such as Cladocopium.
Line 254: I know it’s an illustration, but the CAMAs seen in actual corals are much smaller than those shown in the figure. I wonder if this should be mentioned in the caption. There are TONS of microbes in coral, but the actual amount of SPACE they occupy (e.g., biomass and volume) is small. Check out some of our SEM and TEM images (Peng et al., 2011, Proteomics, Mayfield et al. 2013, Coral Reefs, works of Chii-Shiarng Chen).
Response: We added this information in the figure caption: “The size of the Symbiodiniaceae and microbes has been modified for illustration purposes. For an accurate size representation, see [Peng et al., 2011; Mayfield et al., 2013; and Wada et al., 2019].” We also modified the size of the CAMAs in figure 1, they are now smaller. Please see the attachment.
Lines 343-346: Exactly. This applies to ALL the microbial+bleaching work to date. There are a few phrases elsewhere that imply that viruses or bacteria directly cause bleaching, which, to my knowledge, has long been debunked (e.g., lines 395-396 and reference195, which did not find virus-induced bleaching).
Response: We completely agree with the reviewer comment. We revised the sentences saying otherwise. At new lines 459-462: “Although this study suggests that Vibrio spp. might be specifically linked to coral bleaching, current knowledge makes it impossible to disentangle whether their presence, increase in abundance, and virulence in thermally stressed corals is a consequence of bleaching [183] or plays an active role in the development of bleaching [Zhou et al., 2020]. The statement “trigger bacteria-induced bleaching” was also removed, in addition to “such as extracellular proteases and adhesive genes involved in polyp adhesion in Vibrio species, and toxin P encoding genes participating in the inhibition of photosynthesis”.
Line 384 and elsewhere: I think microbiota should almost universally be “microbiome” in the article.
Response: We decided to make a distinction between the microbiota (non-Symbiodiniaceae microbes) and microbiome, as coral microbiomes often include the dinoflagellate algae in the literature. Here are few examples: van Oppen and Blackall, 2019; Gardner et al., 2019; Morrow et al., 2018; Bourne et al., 2016 (among many others).
Line 384: In the same paragraph, “dinoflagellates” does not need to be capitalized, and there are many English mistakes. I would have this section revised by a native English speaker.
Response: We revised this paragraph according to the suggestions made by a native English speaker.
Line 425: “correlate” is the wrong word. You mean “are associated with” or “have been documented alongside.” This is sort of a chicken and the egg situation; does the stress event lead to a microbial shift that then exacerbates bleaching, or does the bleaching process disrupt the coral-dinoflagellate endosymbiosis?
Response: We modified to “are frequently associated with” as suggested. This is an interesting question, and we thank the reviewer for bringing it up. Please see the detailed response above (in answer to the item (8)).
Line 431: love this idea, and I know Vega-Thurber is trying it. I will, too. Do these microbiome shifts occur BEFORE the downstream effects we can see or IN CONCERT WITH them? Hopefully, it’s the former (or else the diagnostic capacity falls to 0).
Response: We also love this idea, and I think Vega-Thurber is trying this approach at Moorea, where the reefs are monitored frequently each year.
Around lines 450: I like the idea that redundancy could reduce potential bleaching impacts, but in this section are you talking about background dinoflagellate symbiont types or bacteria? It could be both, so just make sure to specify which microbes you are referring to. See my suggestion elsewhere about changing the wording to avoid confusion (since Symbiodiniaceae are technically microbes, as well).
Response: We only refer to the microbiota in this particular section, microbes now refers to non-Symbiodiniaceae as suggested.
Figures
(1) For an article of this length, I would like to see an additional 1-2 figures, even if it is something generic like a before and after picture of a bleached coral. Of course, don’t force it if there is really nothing else that could benefit the article.
Response: As previously mentioned, we included an additional figure of bleached corals and reef landscapes. Please see the attachment.
(2) Figure 2. I like the idea of the figure more than I like the visuals. One issue is the scale. The dinoflagellates occupy 90% or more of the cell volume, yet I worry that readers will think of these circles as cells, in which case the scale is way off. Maybe you could use the polyps drawn elsewhere to show this idea instead. BTW, it should be “shuffling.” The icons in the circles are also too small to read, to where the reader cannot glean the major message you are trying to make. I wonder if, below select circles, you could have a very simple plot that shows the declining number of “good microbes” and the increasing number of “bad” ones, as you go from stage to stage? One line would be ever-increasing (the bad) and another line (colored to match the good microbes) would be ever-decreasing. Again, I like this idea, and I don’t disagree with anything in the figure. I just wonder if you can make it more visually compelling and easier to interpret?
Response: We revised the figure 2 according to the reviewer’ suggestions: we incorporated a coral icon in the background of the circle encompassing the bacteria and Symbiodiniaceae, made the icons within the circle bigger, revised “Shuffling”, incorporated bar plots for each microbiome that show the declining of number of “good microbes” and the increasing number of “bad” ones. We also added this in the figure caption: “Each dotted circle represents the microbiome of a coral at each of the dysbiosis stages described, while the bar plots illustrate the declining number of “good microorganisms” and the increasing number of “bad” ones within the coral microbiome at each of the dysbiosis stages. The number and size of the Symbiodiniaceae and microbes within the coral tissues has been modified for illustration purposes.” Please see the attachment.
(3) Figure 2: Should it be “acclimatized” instead of “adapted” Symbiodiniaceae? An individual cell can, of course, not adapt, but if you mean a cell from a population of symbionts that adapted to life at high temperatures, then “adapted” could still be the right term. But if you mean a symbiont that became stronger on account of, for instance, prior stress exposure, then “acclimatized” would be the correct term. See my treatise elsewhere on the notion that corals can simultaneously evolve and acclimatize. Not sure if this could somehow be incorporated into the figure without making it too convoluted (especially given my prior suggestions), but just a thought.
Response: Here we refer to a better adapted Symbiodiniaceae, meaning that this particular population (in this case a Durusdinuim species) has adapted to a higher temperature compared to other ones. We appreciate that the reviewer offered thoughts on how to address this issue of adapted versus acclimatized Symbiodiniaceae, but unfortunately, we were not able to come up with a better way to illustrate this.
References
(1) The references are inconsistently formatted. Sometimes the article’s title is capitalized in its entirety and sometimes not. I think for Microorganisms, only the first word is capitalized.
Response: We revised and formatted the references to meet Microorganisms criteria.
(2) Species names should be italicized in the references.
Response: Species are now in italic.
All the grammatical errors listed below have been corrected as suggested.
Line 41: “that” “and”
Line 43: “constitute” “are” (constitute implies being made from)
Line 43: “foundation” “foundational”
Line 46: cnidarian host
Line 46: family Symbiodiniaceae
Lines 50-51: remove “continuous” since it is not always the case.
Line 51: Symbiodiniaceae partners “dinoflagellate endosymbionts”
Line 58: host-microbe interaction holobiont
Line 64: are, is, have In the same sentence, “several” “numerous”
Line 67: remove “pulse” since you are also referring to longer-term events.
Line 71: dysbiotic
Line 87: “host’s” “host”
Line 285: Orbicella faveolata, not Montastrea (which is the old name).
Line 328: No need to abbreviate quorum sensing since you never mention it again.
Line 398: Despite (never plural)

Reviewer 2 Report
The manuscript entitled “Defining coral bleaching as a microbial dysbiosis within the coral holobiont” sums up the causes of coral bleaching describing how microbial dysbiosis, occurring by environmental stressors among which the warming heating of ocean waters, can be a crucial player in favoring the switch from the healthy state to the disease state in the coral. In the review, the authors define the characteristics of the coral biology and its relationships with marine microbiota composed by prokaryotes and eukaryotes, viruses, protists and fungi. Finally, several suggestions to prevent microbial dysbiosis able to lead corals to death are well described. The review appears well written and structured answering to the main questions arising from the issue. However, there are some parts which need a correction (English language) and that can be better explained or integrated.
Below you can find a list of my suggestions:
Lines 61-62: Can you describe other abiotic factors, if possible?
Line 103: Scleractinian corals comprise more than 1600 species. Check and correct, please.
Line 111: English punctuation: “The cellular glands, on the other hand, …”
Line 114: English language: “These microhabitats change dramatically (in structure and composition)…”
Line 154: English language: “viruses”
Line 347: English language: “viruses”
Line 355: Bacteriophages are viruses able to infect bacteria. Revise, please.
Author Response
Responses to the Reviewer 2 comments:
The manuscript entitled “Defining coral bleaching as a microbial dysbiosis within the coral holobiont” sums up the causes of coral bleaching describing how microbial dysbiosis, occurring by environmental stressors among which the warming heating of ocean waters, can be a crucial player in favoring the switch from the healthy state to the disease state in the coral. In the review, the authors define the characteristics of the coral biology and its relationships with marine microbiota composed by prokaryotes and eukaryotes, viruses, protists and fungi. Finally, several suggestions to prevent microbial dysbiosis able to lead corals to death are well described. The review appears well written and structured answering to the main questions arising from the issue. However, there are some parts which need a correction (English language) and that can be better explained or integrated.
Response: Thank you for your interest in this review. For your information, the manuscript was revised and proofread by a native English speaker as recommended by reviewer 1.
Lines 61-62: Can you describe other abiotic factors, if possible?
Response: We added “temperature, irradiance, pH, water movement, nutrients, among others” at new lines 80-81.
Line 103: Scleractinian corals comprise more than 1600 species. Check and correct, please.
Response: We verified the literature and most studies are mentioning 1400, 1500, or over 1500 species. We only found one study (Campoy et al. 2020) that says about 1600 species, but the authors do not provide the information about the paper from which this number is from. Consequently, we rather be cautious about this number (that is still in debate) and say “radiated into more than 1500 species” and cite this study (Kitahara et al. 2016).
Kitahara MV, Fukami H, Benzoni F, Huang D (2016) The new systematics of Scleractinia: integrating molecular and morphological evidence. In The Cnidaria, past, present and future. Springer, Cham. pp. 41-59.
All the grammatical errors listed below have been corrected as suggested.
Line 111: English punctuation: “The cellular glands, on the other hand, …”
Line 114: English language: “These microhabitats change dramatically (in structure and composition)…”
Line 154: English language: “viruses”
Line 347: English language: “viruses”
Line 355: Bacteriophages are viruses able to infect bacteria. Revise, please.